# Protective Immunity Elicited by VP1 Chimeric Antigens of Bacterial Ghosts against Hand-Foot-and-Mouth Disease Virus

**DOI:** 10.3390/vaccines8010061

**Published:** 2020-02-01

**Authors:** Saisai Gong, Nan Nan, Yakun Sun, Zhili He, Jiajia Li, Fanghong Chen, Tao Li, Nianzhi Ning, Jianxin Wang, Zhan Li, Deyan Luo, Hui Wang

**Affiliations:** 1State Key Laboratory of Pathogen and Biosecurity, Beijing Institute of Microbiology and Epidemiology, 20 Dongda Street, Fengtai District, Beijing 100071, China; gby1738131995@163.com (S.G.); nannan163n@163.com (N.N.); syk_dyx@hotmail.com (Y.S.); lili892017173@163.com (Z.H.); 13024096923@163.com (J.L.); cfh880314@163.com (F.C.); litaobmi@126.com (T.L.); ningnianzhi@163.com (N.N.); jianxinwang1994@163.com (J.W.); yexi19881214@126.com (Z.L.); 2Department of Microbiology, Anhui Medical University, Hefei 230000, China

**Keywords:** bacterial ghost, hand, foot, and mouth disease, enterovirus 71, Coxsackie virus

## Abstract

This study was designed to evaluate the immunogenicity and protective efficacy of two VP1 chimeric antigens of bacterial ghosts. Inoculation of the two VP1 chimeric antigens of bacterial ghosts into BALB/c mice markedly elicited humoral and mucosal immune responses. The specific antibodies induced by the chimeric ghosts protected mice not only against the virus that causes hand-foot-and-mouth disease but also against *E. coli* O157:H7 bacterial infection. In comparison with the negative control, immunization with the chimeric ghosts protected mice against two LD_50_ hand-foot-and-mouth disease viral infection. In addition, this specific immunity also protected the pups of pregnant mice immunized with the VP1 chimeric antigens of bacterial ghosts against 20 MLD *E. coli O157:H7* infection. Taken together, the results of this study verify for the first time that the VP1 chimeric antigens of bacterial ghosts are target candidates for a new type of vaccine against hand-foot-and-mouth disease. Additionally, this vaccine strategy also elicited a stronger immune response against *E. coli* O157:H7.

## 1. Introduction

Hand-foot-and-mouth disease (HFMD), a global infectious disease caused by intestinal viruses, has differing outbreak intensities all over the world [1]. Similar to other intestinal pathogens, host resistance to the HFMD virus relies mainly on innate and mucosal immunity [2,3]. The mature B lymphocytes that secrete serum-neutralization antibodies is the most important protective response in infected hosts [4]. A vaccine that is nonpathogenic to humans, but effective in stimulating a broad protective immune response, is needed to control HFMD. To develop this type of HFMD vaccine, several research groups are pursuing different strategies, including the development of subunit vaccines and overexpression of protective homologous antigens [5,6,7,8]. Another new strategy for developing safe and efficacious vaccines is immunization with bacterial ghosts (BGs) carrying a protective antigen. BGs are produced by the expression of PhiX174 lysis gene *E* and result in cytoplasmic loss and cellular lysis [9]. BGs maintain the cellular morphology and native antigenic surface structure [10] and possess the adjuvant property of bacteria [11,12]. The effects of BG vaccines have been described in various pathogens, such as *Vibriocholera* [13], *Helicobacterpylori* [14], and *Escherichia coli* O157:H7 [15].

HFMD is primarily caused by enterovirus 71 and the Coxsackie virus, both of which are in the *Enterovirus* genus. Studies have shown that using the VP1 capsid protein of the two viruses as an antigen provides protective immunity against viral infections in a murine model [6,16]. In this study, linear VP1 of the enterovirus 71 (EVP1) and the Coxsackie virus (CVP1) were displayed on the surface of *E. coli* O157:H7 BGs based on the sandwich vector pSOmpA [17]. The outer membrane protein A (OmpA) of *E. coli* was used in order to construct a novel candidate vaccine named EVP1 bacterial ghosts (EBGs) and CVP1 bacterial ghosts (CBGs). The immunogenicity, protective ability, and immunologic mechanism of these vaccines in the challenge of the HFMD virus and enterohemorrhagic *E. coli* O157:H788321 (EHEC) were analyzed in this study.

## 2. Materials and Methods 

### 2.1. Bacterial Strains, Cells, Plasmids and Virus

Two *E. coli* strains were grown in Luria–Bertani (LB) broth or agar (Oxoid LTD, Basingstok, Hampshire, England) supplemented with 100 μg/mL of ampicillin for selection of recombinant plasmid at 37 °C. The bacterial strain *E. coli* O157:EDL 933 for bacterial ghosts preparation was kept in our lab. The bacterial strain *E. coli* O157:H788321 for challenge was also kept in our lab. Two wild-type strains ATCC O157:H788321 were purchased from the ATCC center (American Type Culture Collection). A Vero E6 cell line and a HEp-2 cell line were kept in our lab. The expression vector pGEX was purchased from Transgen Inc. (Beijing, China), and display vector pSOMPA and lysis plasmid pLysisE were constructed by our lab [15]. Enterovirus 71 (EV71; GenBank: JQ514785.1) and coxsackievirus B3 (CB3; GenBank: M88483.1), on the basis of which vaccines have been developed, were kept in our lab. 

### 2.2. Mice 

All animal studies were conducted in accordance with the Beijing Institute of Microbiology and Epidemiology Animal Care and Use Committee guidelines (IACUC 2012). BALB/c wild type mice (5-week-old, weighing 14–16 g) were obtained from our institute’s Laboratory Animal Center, Beijing, China. All experimental mice were bred in a specific pathogen-free facility at our institute. Experimental mice were matched for age and sex and cared for according to the guidelines of our institute. Mice were monitored and weighed at least once daily after initiating infection. Recumbent mice, and mice that lost more than 30% weight, were considered moribund and euthanized. 

### 2.3. Construction and Preparation of pOEVP1 and pOCVP1

Full-length open reading frames of the VP1 genes from enterovirus 71 (EVP1) and coxsackievirus B3 (CVP1) were amplified with PCR. The 234–325 amino acids of OmpA amplified from *E. coli* O157:H7 EDL 933 strain. The PCR primers were designed as follows: EVP1, Forward, 5’-GAATTCGGAGATAGGGTGGC-3’, Reverse, 5’- GAGCTCAAGAGTGGTGATCG-3’. CVP1 Forward, 5’- GAATTCGGCCCAGTGGAAGAC-3’, Reverse, 5’- GAGCTCAAATGCGCCCGTAT-3’. The two fragments EVP1 and OmpA were digested by *Eco*RI/*Sac*I (New England Biolabs, Ipswich, MA, USA) and then ligated into the expression vector pGEX between *Xma*I and *Nco*I restriction sites to generate the expression plasmid pOEVP1. The two fragments CVP1 and OmpA were digested by *Eco*RI/*Sac*I and then ligated into the expression vector pGEX between *Xma*I and *Nco*I restriction sites to generate the expression plasmid pOCVP1. The resulting constructs were transformed into *E. coli* O157:H7 EDL 933 for bacterial ghosts preparation.

### 2.4. Preparation of Bacterial Ghosts and Whole Cells [18]

The pOEVP1 and pOCVP1 plasmids were transformed into *E. coli* O157:H7EDL 933 as novel competent cells. LB-medium containing ampicillin was inoculated with *E. coli* O157:H7 EDL 933 overnight culture, transformed with the kanamycin resistance and thermolysis plasmid pLysisE to generate BGs (named EBGs and CBGs). *E. coli* O157:H7 EDL933 strain without pOEVP1 or pOCVP1 plasmid were prepared to BGs as control (named OBGs). In detail, 200 ml of LB-medium containing 100 µg/mL ampicillin and 100 µg/mL kanamycin was inoculated with 5 mL of each *E. coli* strain containing plasmids. Grown up to OD_600_ 0.3, the cultures of EBGs and CBGs were induced by isopropyl β-D-thiogalactopyranoside (IPTG) at a final concentration of 1 mM at 28 °C. The induction of lysis was achieved by shifting the temperature from 28 to 42 °C when the OD_600_ reached 0.6, and the procedure was monitored by the optical densities. The lysis rate based on colony-forming units (CFU) and the morphous of EBGs and CBGs were detected as described previously [19]. Subsequently, BGs vaccine candidates were prepared by the repeated freezing and thawing method to get rid of the surviving bacteria, and then centrifuged 10 min and washed twice in ice-cold buffer (25 mM TBS, Tris-Buffered Saline). These steps may need to be repeated one more time. For long term storage, bacterial ghosts should be lyophilized. The absence of viable cells in the lyophilized samples was determined using colony-forming units. 

### 2.5. Analysis of Antigens by Fluorescence Activating Cell Sorter (FACS )

Phosphate buffered solution (PBS) containing mouse anti-EVP1/CVP1 immunoglobin G was incubated with EBGs and CBGs for 1 h, respectively. Then, the FITC-labeled goat anti-mouse IgG was added and incubated for another 1 h as a second antibody, according to the first antibody, respectively. After processing, the labeled EBGs and CBGs were incubated with 10^6^ HEp-2 cells for 4 h. Finally, these cells were harvested and re-suspended with FACS buffer, and 1 × 10^5^ HEp-2 cells were detected by flow cytometry. The OBGs cells were used as control.

### 2.6. Analysis of Cytotoxicity by MTT(3-(4,5-dimethyl-2-thiazolyl)-2,5-diphenyl-2-H-tetrazolium bromide)

Cytotoxicity assays of lyophilized EBGs and CBGs were performed by MTT assay on Vero E6 cells. Serial diluted (1:10) EBGs and CBGs were applied on a Vero E6 cell monolayer and incubated at 37 °C (CO_2_) for 36 h. Then, 100 μL/well 0.5 mg/ml 3-(4,5-Dimethylthiazol-2-yl)-2,5-diphenyl tetrazolium bromide were added and incubated for another 4 h to produced formazan crystals. The produced formazan crystals were re-suspended by dimethyl sulphoxide (DMSO, 150 μL/well), and the absorbance at 570 nm was detected. The OBGs and *E. coli* O157:H7EDL 933 were used as controls.

### 2.7. Mice Immunization and Challenge

All animal studies were conducted in accordance with the Beijing Institute of Microbiology and Epidemiology Animal Care and Use Committee guidelines. Three-week-old male and female BALB/c mice were obtained from our institute’s Laboratory Animal Center and divided into groups randomly (20 mice/group). All experimental mice were bred in a specific pathogen-free facility at our institute. Experimental mice were matched for age and sex and cared for according to the guidelines of the institute. The mice were intragastrically administrated with 0.1 mg (corresponding to 10^8^ CFU) of EBGs and CBGs, which were resuspended in 100 μl of PBS on day 0 for the primary injection and day 14 for boost vaccinations. PBS alone was administrated to the vehicle control group. Twelve mice from each group were then challenged intragastrically 14 days after booster with 20 LD_50_ (2 × 10^9^ CFU) or 50 LD_50_ (5 × 10^9^ CFU) of viable *E. coli* O157:H788321 strain. Adult mice were resistant to EV71 (or CB3) administrations.

Eight mice from each group were separated and bred for the subsequent study. The pups were used for a virus challenge study. The experimental endpoint was determined, as previously described [20]. One-day-old mice were used to examine the role of maternal antibodies in challenge studies. For maternal immunization, the BALB/c suckling mice possessing passively-transferred maternal EV71 or CB3 antibodies were challenged 1 day after birth with the virus of EV71 and CB3. The mice were observed daily for the occurrence of mortality as the experimental endpoint.

### 2.8. Detection of Specific Antibodies 

The IgA/IgG of serum and irrigating solution specific to OBGs and proteins (intimin, EVP1, and protein CVP1) were measured by enzyme-linked immunosorbent assay. The presence of serum IgG and of the subtypes IgG1, IgG2a, IgG2b, and IgG3 specific to vaccine candidates was determined by indirect ELISA. The BGs (EBGs and CBGs, 0.1 mg/mL) and purified proteins (intimin, EVP1, and CVP1, 0.1 mg/mL) were coated in 96-well plates overnight at 4 °C, respectively. Serum samples were serially diluted in 2-fold dilutions from 1:10 to 1:20480. The endpoint dilution titer was calculated as the serum dilution resulting in an absorbance reading of 0.2 units above background. Goat anti-mouse IgA-HRP (Sigma, 1:5000) or goat anti-mouse IgG-HRP (Sigma, 1:5000) were used as the detection antibodies. The reactions were developed with TMB (3,3’,5,5’-Tetramethylbenzidine) and stopped with 2 M H_2_SO_4_. The absorbance at 450 nm was detected.

### 2.9. Statistical Analysis

Statistical analyses were performed using the program Prism 5.0 (GraphPad Software, Inc., LaJolla, California, CA, USA). Values are expressed as mean ± SD. Data were analyzed by unpaired Student’s *t-*test (normal distribution) or one-way ANOVA followed by Dunnett’s multiple comparison test. Survival data were analyzed by log-rank tests. Values of *p* < 0.05 are considered to be statistically significant.

## 3. Results

### 3.1. Preparation and Evaluation of the Chimeric BG Vaccine Candidates

We constructed specific BG vaccine candidates, EBG and CBG, in order to compare the relative roles of these BGs vaccine candidates in inducing an immune response and protective immunity against HFMD virus infection. The expression plasmid pOEVP1 containing 1990 bp of the *OEVP1* gene was constructed, and the co-transformation of plasmids pOEVP1 and pLysE successfully generated the vaccine strain of EBGs. The expression plasmid pOCVP1 containing 1950 bp of the *OCVP1* gene was constructed, and the co-transformation of plasmids pOCVP1 and p LysE successfully generated the vaccine strain of CBGs. The OD_600_ of both EBGs and CBGs was reduced constantly after a shift in temperature. The precipitate 1 h after induction was harvested in order to evaluate the lysis rate, which was counted as 99.99% ± 0.01% (Figure 1A,B). The morphology of EBGs and CBGs was detected by electron microscopy. Electron microscopy studies showed that the protein VP1-specific transmembrane tunnel structure was not randomly distributed over but was restricted to areas of division sites, predominantly in the middle of the cell or at polar sites [9]. Electron micrographs of a typical bacterial ghost are presented in Figure 1D. To evaluate the safety of the two vaccine candidates, we treated Vero E6 cells with 10 mg of chimeric BGs (equivalent to 1 × 10^10^ CFU of bacteria) and 1 × 10^7^ CFU of *E. coli* O157:H7. We found that the pathogenic bacteria killed nearly 100% of the Vero E6 cells. In contrast, no obvious cytotoxic effects were detected when the Vero E6 cells were treated with 10 mg of EBGs or CBGs (*p* > 0.05; Figure 1C). The outer membrane protein intimin was confirmed in EBGs, CBGs, and OBGs by flow cytometry, but the EVP1 and CVP1 were detected only on the surfaces of EBGs and CBGs (Figure 1E).

### 3.2. The Mucosal Immune Response Elicited by Immunization with EBGs or CBGs

In order to investigate the mucosal immune response induced by the various BGs vaccines, we analyzed the IgA antibody titers specific to BGs, intimin, and VP1 proteins in the serum and irrigating solution of the intestinal tract. Both the sera and irrigating solutions were collected on days 0, 7, 14, 21, and 28 (3 mice were used for each time point) after the last immunization and were assayed for the presence of IgA antibodies via (enzyme linked immunosorbent assay) ELISA. The results showed that the IgA titers specific to BGs in the sera and irrigating solutions of mice immunized with EBGs and CBGs increased at day 14 and peaked on day 28 (Figure 2). The IgA titers specific to EBGs or CBGs in the hyperimmune sera of immunized mice reached 1:160 and 1:130, respectively (Figure 2A). The IgA titers specific to EBGs or CBGs in the irrigating solutions of immunized mice reached 1:260 and 1:210, respectively (Figure 2B). The IgA titers specific to intimin in the hyperimmune sera of immunized mice reached 1:100 and 1:130, respectively (Figure 2C). The IgA titers specific to intimin in the irrigating solutions of immunized mice reached 1:130 and 1:210 (Figure 2D). The IgA titers specific to the EVP1 protein in the hyper immune sera of mice immunized with EBGs reached 1:130, and no antibody titers were detected in mice immunized with CBGs (Figure 2E). The IgA titers specific to the EVP1 protein in the irrigating solutions of mice immunized with EBGs reached 1:60, and no antibody titers were detected in mice immunized with CBGs (Figure 2F). The IgA titers specific to the CVP1 protein in the hyperimmune sera of mice immunized with CBGs reached 1:60, and no antibody titers were detected in mice immunized with EBGs (Figure 2G). The IgA titers specific to the CVP1 protein in the irrigating solutions of mice immunized with CBGs reached 1:130, and no antibody titers were detected in mice immunized with CBGs (Figure 2H).

### 3.3. The Humoral Immune Response against HFMD Virus Elicited by Immunization with EBGs or CBGs

In order to further investigate the humoral immune response induced by the various BG vaccines, we analyzed the IgG antibody titers specific to BGs, intimin, and VP1 proteins in the serum and irrigating solution of the intestinal tract. Samples were collected on days 0, 7, 14, 21, and 28 (3 mice were analyzed at each time point after the last immunization and were assayed for the presence of EVP1 and CVP1-specific antibodies by ELISA. Similar to our previous report [15], the specific antibodies induced by CBGs and EBGs increased persistently over time. We detected high levels of specific anti-EBGs and anti-CBGs IgG antibodies in the samples collected from mice immunized with BGs vaccine candidates. No specific antibodies were detected in the PBS-immunized group (Figure 3A,B). The antibodies specific to intimin, one of the most important factors involved in the adhesion of EHEC, were tested. In contrast to PBS, both the CBGs and EBGs were able to induce significant intimin-specific IgA/IgG antibodies (Figure 3C,D). The IgG titers specific to the EVP1 protein in the sera of mice immunized with EBGs reached 1:130, and no antibody titers were detected in mice immunized with CBGs (Figure 3E). We did not detect specific IgG antibodies in the irrigating solutions from mice immunized with EBGs and CBGs (Figure 3F). The IgG titers specific to the CVP1 protein in the sera of mice immunized with CBGs reached 1:210, and no antibody titers were detected in mice immunized with EBGs (Figure 3G). We did not detect specific IgG antibodies in the irrigating solutions from mice immunized with EBGs and CBGs (Figure 3H). The analysis of IgG subtypes showed a significant increase in IgG1 in the serum of mice immunized with EBGs or CBGs. The titers reached over 1:1000. The titers of IgG2a and IgG2b were also detected, but only reached about 1:200 (Figure 4).

### 3.4. Efficacy of EBGs and CBGs Immunization in Generating Protective Immunity against the HFMD Virus

While the vaccine approach has great potential for HFMD prevention, immunoprotection against HFMD has not been achieved. In order to study the protective effect of EBGs and CBGs, pups were challenged with an intraperitoneal injection of the EV71 and CB3 viruses. The results showed that EBGs protected mice against 2 LD_50_ of EV71 and CB3 infections (Figure 5A,B). When mice were challenged with 5 LD_50_ EV71, 50% of the suckling mice immunized with EBGs survived; however, none of the suckling mice immunized with CBGs survived. CBGs provided an 87.5% protective rate against 2 LD_50_ CB3 infection and 37.5% protective rate against 5 LD_50_ CB3 infection, whereas there were no surviving mice in the 2 LD_50_ EBGs-immunized group for either challenge. There were no survivors in control groups.

### 3.5. Efficacy of EBGs and CBGs Immunizations in Generating Cross-Protective Immunity against E. coli O157:H788321

EBGs and CBGs not only displayed stronger protection against EV71 and CB3 infections but also displayed cross-protection against *E. coli* O157:H788321 infection. EBGs provided a similar (*p* > 0.05) protective rate (14/20, 70%) as the CBGs (15/20, 75%) when the challenge dose was 20 LD_50_ (Figure 5C). When the intragastric challenge dose was increased to 50 LD_50_, there were no survivors in either immunized group (data not shown). No mice survived in the PBS-immunized groups when challenged with either the high or low dose.

## 4. Discussion

In recent years, there has been an increase in the occurrence of HFMD, which has no effective treatment. Disease prevention is a promising approach to defend against HFMD virus infection, which deserves further exploration [16]. As mentioned previously, capsid protein VP1 is a major virulence determinant of the HFMD virus. A number of studies have demonstrated that antibodies induced by VP1-based vaccines can efficiently block viral replication in tissue [21,22]. Although these VP1 protein vaccine candidates can be produced easily, they are less immunogenic and require a delivery system or the addition of adjuvants. BGs are produced by the expression of the cloned PhiX174 gene E in Gram-negative bacteria [23], which can be used as a delivery system for foreign subunit antigens. It has been demonstrated that foreign antigens presented by BGs to the immune system can induce strong immune responses [14,24,25].

Therefore, we designed two VP1 chimeric antigens of BGs as vaccine candidates in this study. We immunized mice with the two chimeric BGs and challenged them with *E. coli* O157:H7. We also challenged suckling mice with EV71 and CB3 virus separately. In this study, it was verified for the first time that two BG vaccines based on the VP1 proteins induced significant mucosal and humoral immunity. We detected high levels of specific IgG and IgA titers 7 days after the first immunization, and the titers were much higher on day 7 after a booster. From these results, we concluded that both CBGs and EBGs were highly immunogenic for mice. To investigate which subtype was predominant in mice immunized with the vaccine candidates, the IgG subtypes were assayed by ELISA. We found a significant increase in the absorbance for antigen-specific IgG1, IgG2a, and IgG2b, but not IgG3.

Recombinant VP1 protein formulated with complete Freund’s adjuvant can elicit neutralizing antibody responses in mice. Synthetic immunogens based on the neutralization epitopes of EV71 also require complete Freund’s adjuvant, which is not acceptable for use in humans [16]. Vaccination with DNA plasmid constructs encoding VP1 resulted in low neutralizing responses. Vaccination of neonatal mice with an adenovirus vector expressing a conserved neutralization epitope conferred protection against lethal EV71 challenge. Thus, delivery systems are very important for HFMD virus vaccine development. Here, we selected BGs as a delivery system for our vaccine candidates. In order to evaluate the protection conferred by the vaccine candidates, the newborn mice were challenged with an intraperitoneal injection of EV71 and CB3 virus separately. We found that the two vaccine candidates protected mice against HFVD virus infection.

The immunized adult mice were intragastrically challenged with *E. coli* O157:H7. We found that the two vaccine candidates also protected mice against *E. coli* infection. Intimin protein, a membrane component on the surface of *E. coli* O157:H7, is an important factor for bacterial attaching/effacing (A/E), and the levels of intimin specific antibodies have been demonstrated to be associated with protection against this bacterium [26,27]. In our study, we evaluated high levels of anti-intimin antibody responses. Therefore, the novel BG vaccine candidates have the potential to provide protection against the adhesion of bacteria. We did not find a significant difference in the protection of CBG-and EBG-immunized mice challenged with *E. coli.*

To summarize, our chimeric BGs vaccine candidates elicit a higher mucosal immune response and provide greater protection for the host against HFMD. Our data decisively demonstrated that our vaccine candidates also conferred cross-protection against *E. coli* O157:H7, indicating that the BGs can be used as a relatively efficacious vector for vaccine development against HFMD.

## Figures and Tables

**Figure 1 vaccines-08-00061-f001:**
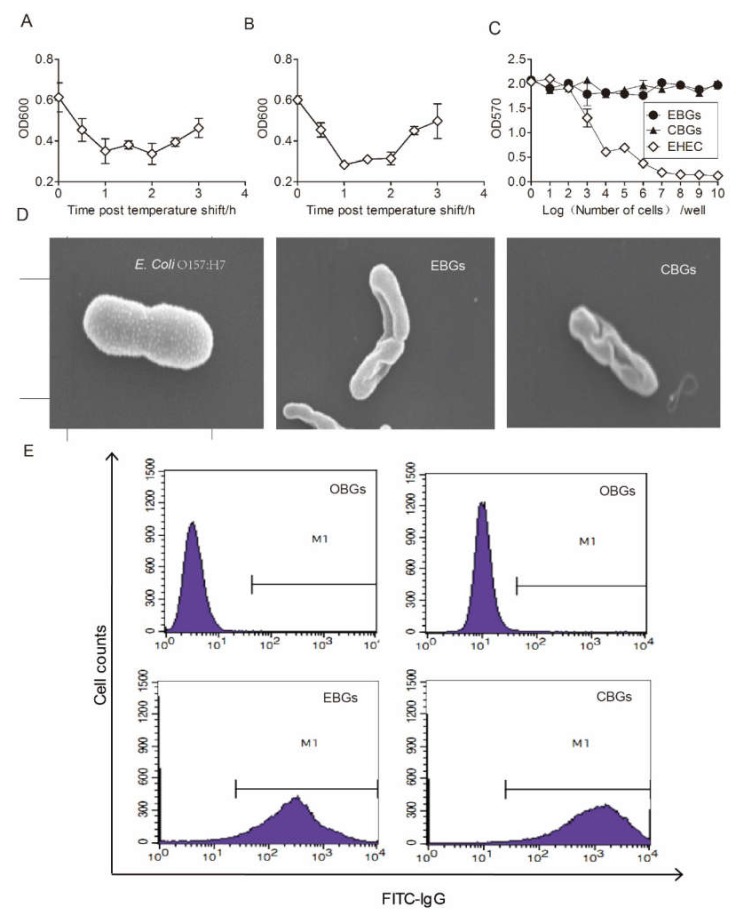
Verification of the bacterial ghost (BG) vaccine candidates and evaluation of the cytotoxicity. (**A**) Pyrolysisrates of EVP1 bacterial ghosts (EBGs). (**B**) Pyrolysisrates of EBGs. The OD_600_ of both EBGs and CVP1 bacterial ghosts (CBGs) were reduced after the shift in temperature. (**C**) Cytotoxicity analysis of BGs. Serially diluted BGs were incubated with a monolayer of Vero E6 cells for 36 h, and the MTT assay was performed to detect cytotoxicity. (**D**) The morphology of EBGs and CBGs was assessed by electron microscopy (Scale: *E. coli*, 12.9 mm × 25.K SE; EBGs, 12.9 mm × 18.K SE; CBGs, 12.9 mm × 37.K SE). (**E**) Verification of the surface antigens. Flowcytometry was used to detect 1 × 10^5^ HEp-2 cells. Analysis of *E. coli* O157:H7 chimeric BG outer membrane proteins, which were identified using an EVP1 or CVP1 antibody.

**Figure 2 vaccines-08-00061-f002:**
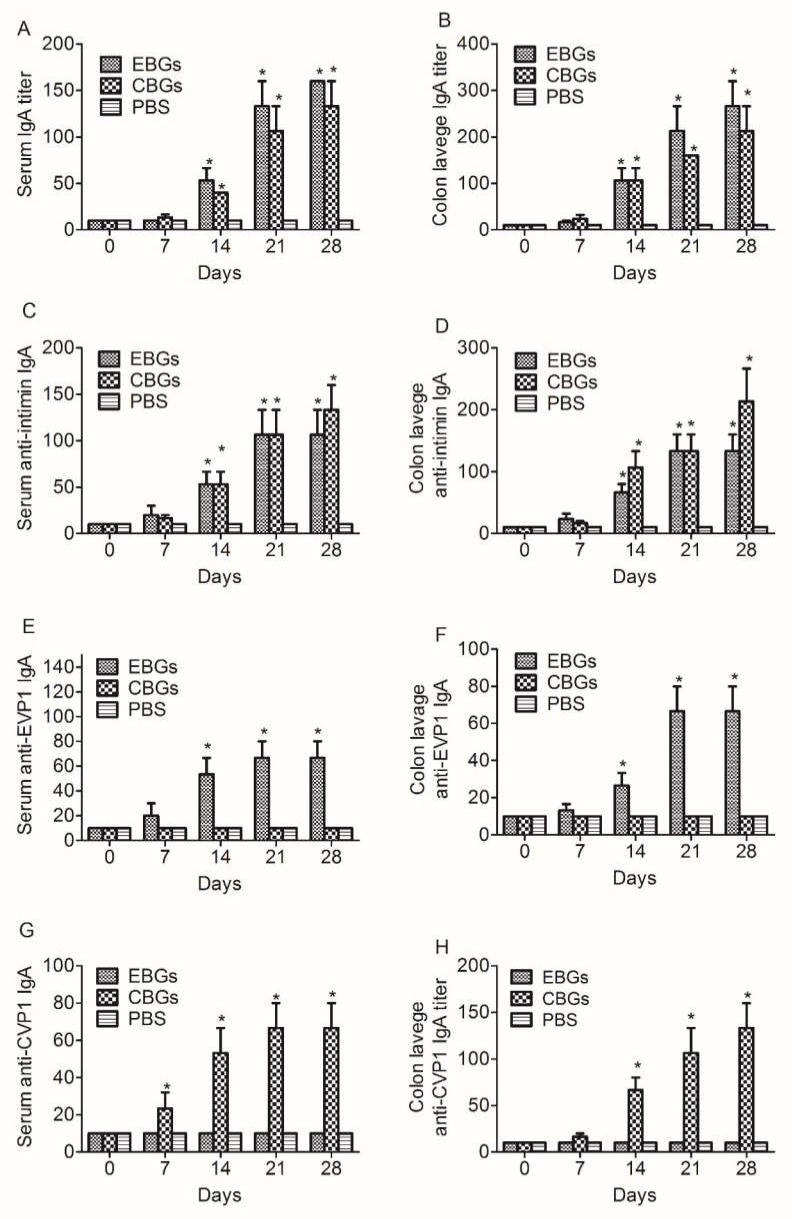
Detection of specific IgA titers in the sera and irrigating solution of immunized mice. Mice were inoculated intragastrically with BGs. Mice that received a PBS injection were negative controls. The IgA levels in the serum and irrigating solution specific to OBGs (*E. coli* O157:H7 EDL933 strain without pOEVP1 or pOCVP1 plasmids) were measured for specific proteins (intimin, EVP1, and protein CVP1) by enzyme-linked immunosorbent assays. (**A**) Time course of serum IgA titers specific to OBGs. (**B**) Time course of irrigating solution IgA titers specific to OBGs. (**C**) Time course of serum IgA titers specific to intimin. (**D**) Time course of irrigating solution IgA titers specific to intimin. (**E**) Time course of serum IgA titers specific to protein EVP1. (**F**) Time course of irrigating solution IgA titers specific to protein EVP1. (**G**) Time course of serum IgA titers specific to protein CVP1. (**H**) Time course of irrigating solution IgA titers specific to protein CVP1. Data are from two independent experiments. * *p* < 0.01, ** *p* < 0.001.

**Figure 3 vaccines-08-00061-f003:**
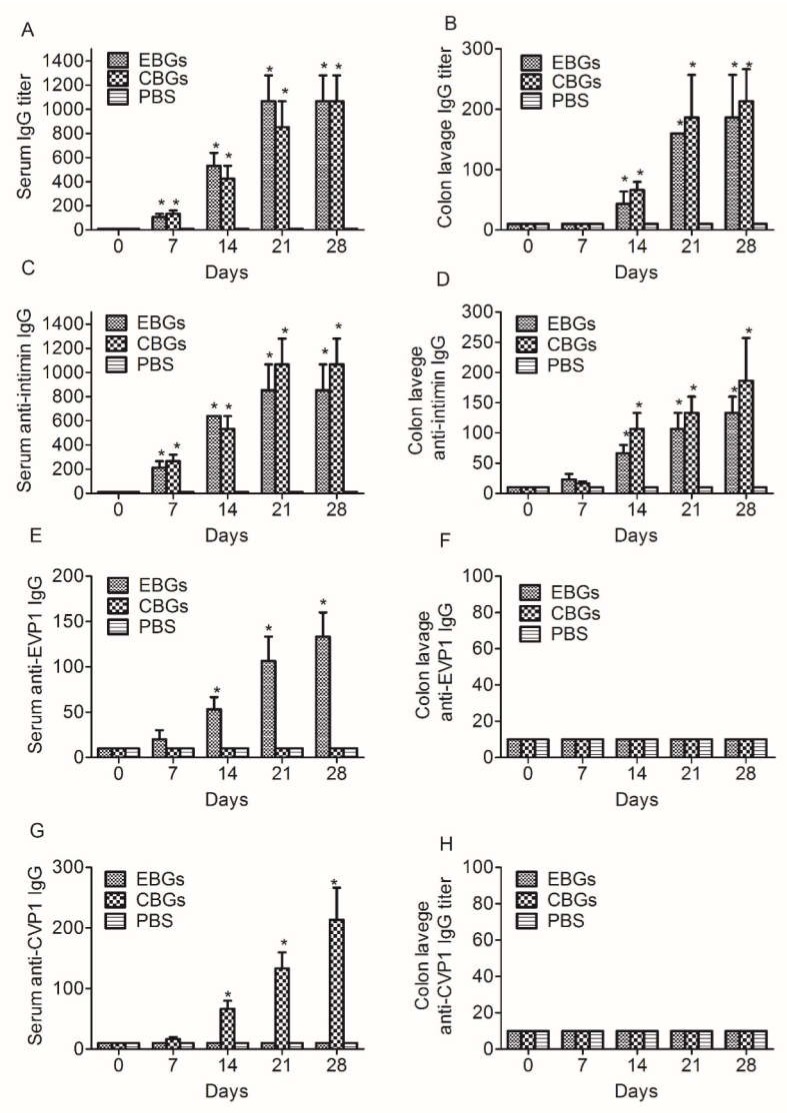
Detection of the specific IgG titers in the sera and irrigating solution of immunized mice. Mice were inoculated intragastrically with BGs. Mice that received a PBS injection were negative controls. The IgG of serum and irrigating solution specific to OBGs and proteins (intimin, EVP1, and protein CVP1) were measured by enzyme-linked immunosorbent assays. (**A**) Time course of serum IgG titers specific to OBGs. (**B**) Time course of irrigating solution IgG titers specific to OBGs. (**C**) Time course of serum IgG titers specific to intimin. (**D**) Time course of irrigating solution IgG titers specific to intimin. (**E**) Time course of serum IgG titers specific to protein EVP1. (**F**) Time course of irrigating solution IgG titers specific to protein EVP1. (**G**) Time course of serum IgG titers specific to protein CVP1. (**H**) Time course of irrigating solution IgG titers specific to protein CVP1. Data are from two independent experiments. * *p* < 0.01.

**Figure 4 vaccines-08-00061-f004:**
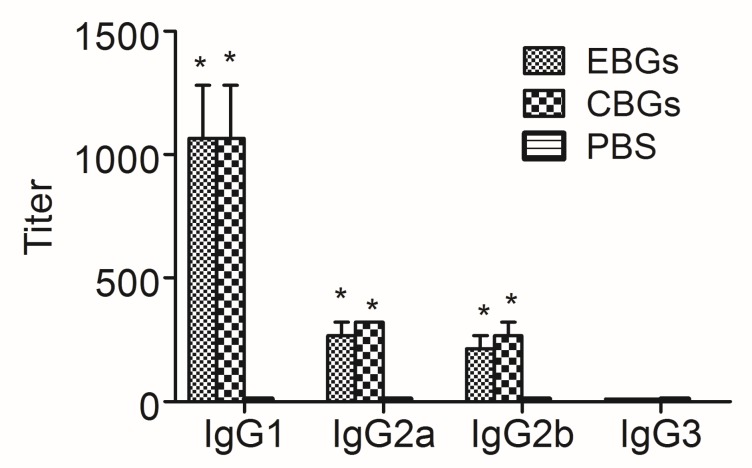
Antibody subtype profiles of mice immunized with various vaccines. Mice were inoculated intragastrically with BGs. Mice that received a PBS injection were negative controls. Two weeks after the last immunization, both the sera and irrigating solution were collected from the experimental mice, and antibody titers were evaluated by an ELISA. Data are from two independent experiments. Each bar represents the mean titers of antibodies in the same group. * *p* < 0.01, as compared with the PBS group.

**Figure 5 vaccines-08-00061-f005:**
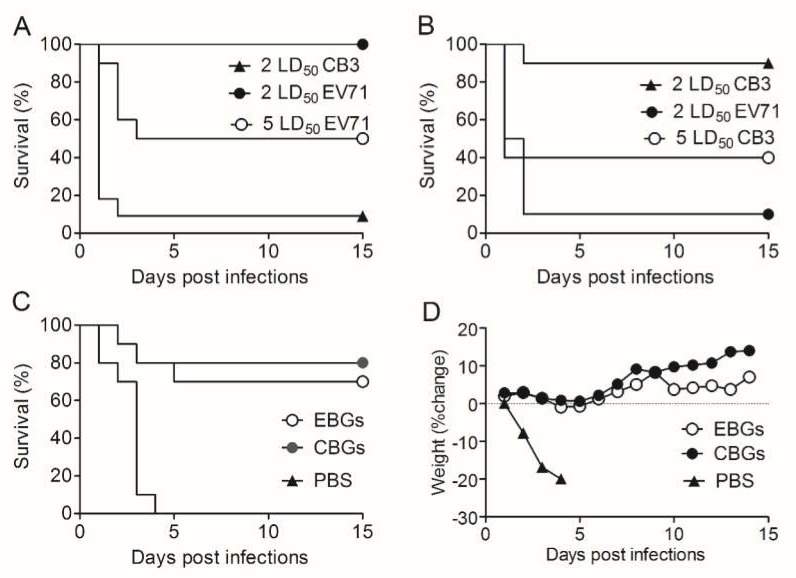
Vaccine candidates protected mice against a lethal dose of toxins. (**A**) Survival rates of the pups of mice immunized with BGs and challenged with 2 LD_50_ CB3, 2 LD_50_ EV71, or 5 LD_50_ EV71. In comparison with mice immunized with PBS, all vaccine candidates increased the survival rate (*p* < 0.001; *n* = 20 mice/group). (**B**) Survival rates for the pups of mice immunized with BGs and challenged with 2 LD_50_ EV71, 2 LD_50_ CB3, or 5 LD_50_ CB3. In comparison with mice immunized with PBS, all vaccine candidates increased the survival rate (*p* < 0.001; *n* = 20 mice/group). (**C**) Survival rate (**D**) body weight changes for wild-type mice immunized with BGs and challenged with 20 LD_50_ (2 × 10^9^ CFU) of viable *E. coli* O157:H788321 strain. In comparison with mice immunized with PBS, all vaccine candidates increased the survival rate (*p* < 0.001; *n* = 20 mice/group). Similar results were observed in two independent experiments. Survival data were analyzed by log-rank tests. Data are from two independent experiments.

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
