# Peer review of "Protective Immunity Elicited by VP1 Chimeric Antigens of Bacterial Ghosts against Hand-Foot-and-Mouth Disease Virus"

_vaccines, 2020, doi:10.3390/vaccines8010061_

Round 1

Reviewer 1 Report

This manuscript described a GB based vaccine against HFMD (and E. coli). Overall, the study is well designed and the data is well presented. Here are some comments:

General comments:

Does the author measure the LPS level in these vaccines? Is there any endotoxic effect observed in the vaccinated mouse? Is there any other clinic sign of HFMD disease observed in the challenged group (vaccinated and control group)? Could the author provide the temperature/body weight data? Why do you choose CB3 instead of CA16 for the vaccine, since the CA16 is more clinical significant?

Specific comments:

Some Misspelling vocabulary needs to be corrected. In line 33, “A vaccine that is noninfectious to humans…”, the word “noninfectious” might be changed to nonpathogenic or avirulent, etc. Does the protein/vaccine of VP1 contains GST tag, since the manuscript does not mention the cleavage/cut procedure either?  In line 98-100, is mouse anti-VP1 first antibody, and labeled rabbit anti-chicken, goat anti-rabbit second antibody?

Author Response

Dear reviewer:

We thank your good comments and believe that we have addressed all of your concerns.

Sincerely, 

Hui Wang

                  Response to Reviewer 1 Comments

General comments:

 1. You expressed your concern about “Does the author measure the LPS level in these vaccines? Is there any endotoxic effect observed in the vaccinated mouse?”

Response 1: Sorry, we did not measure the LPS levels for this animal research model. We did not observe any endotoxic effect in vaccinated mouse. It is a really nice question; it is necessary to measure the LPS levels when we start to research on human used vaccine.

2. You expressed your concern about “Is there any other clinic sign of HFMD disease observed in the challenged group (vaccinated and control group)? Could the author provide the temperature/body weight data?”

Response 2: Yes, the mice began to model a typical dull hair, hair shaft phenomenon and lost weight after challenge adult mice in control group. It is hard to monitor the temperature. Most of pups died before day 3, so these tiny mice look too weak to weight every day. We added body weight changes data about adult mice to Figure 5.

3.You expressed your concern about “Why do you choose CB3 instead of CA16 for the vaccine, since the CA16 is more clinical significant?”

Response 3: This is a really nice question. We choose CB3 because it is one of pandemic strain in China, and also it is a virulent strain which we kept.

Specific comments:

1.You expressed your concern about “Some Misspelling vocabulary needs to be corrected. In line 33, “A vaccine that is noninfectious to humans…”, the word “noninfectious” might be changed to nonpathogenic or avirulent, etc.”

Response 1: Thank you for you good suggestions. We corrected this word. Please see line 33.

2.You expressed your concern about “Does the protein/vaccine of VP1 contains GST tag, since the manuscript does not mention the cleavage/cut procedure either?”

Response 2: No, this vaccine does not contain GST tag for cleavage. The VP1 proteins were expressed on the surface of bacteria to make the EBG or CBG. We added more details about BGs preparations, please see the method 2.4.

3.You expressed your concern about “In line 98-100, is mouse anti-VP1 first antibody, and labeled rabbit anti-chicken, goat anti-rabbit second antibody?”

Response 3: Sorry, I made a mistake in writing this section because most situation we used these for second antibody in our lab. Please see reference 15. This time, we changed. We checked records and corrected, please see line 119-120.

Reviewer 2 Report

Gong and colleagues tested the immunogenicity and efficacy of a bacterial ghost cell vaccine containing enterovirus VP1 antigens for their ability to protect mice against HFMD virus challenge and E. coli challenge.  Unfortunately, Fig 1A demonstrates that a significant portion of bacteria in culture were not lysed following temperature shift.  Consequently, the material that was later lyophilized must consist of bacterial ghosts AND cells that were viable and proliferating when the culture was harvested.  The methods are not well described.  The authors state that no live bacteria were detected after lyophilization, but experimental details aren't clear about whether or not the mice were immunized with lyophilized vaccine or some other form.  This is a major flaw in the manuscript.  Additionally, the ELISA method is not sufficiently described, so it's impossible to fully interpret significant portions of the paper.  Please see my detailed response below.

Major Comments:

Lines 31-33. This sentence needs to be significantly revised.  What B cells secrete “serum-neutralization antibodies into the intestine”?  Additionally, the sentence confuses protective immunity with the effector immune response.  FMDV obviously generates and disseminated infection and is not restricted to the mucosa.  The immune mechanisms for clearing that infection from an infected individual are not necessarily the same as the correlate of protection against infection.  Lastly, the paper listed as a reference for this sentence has nothing to do with FMDV. The authors often omit the space between a number and the word that follows it, and frequently between words. I’ve noted several instances below, but the paper deserves a careful review to fix all of these minor issues.  None of them alone is particularly noteworthy, but the frequency is great enough that it’s distracting.  The axes on 1A read “time post inducing” but I think this is really the time post temperature shift as stated in the legend. See also lines 89-92 in methods, which state that the cultures were induced at OD600=0.3 and temperature shifted at OD600=0.6.  The increases in OD600 seen after two hours is extremely concerning.  Do viable bacteria remain in the culture after induction of lysis?  Line 96 states that lyophilized samples were tested for viable cells, but based on this figure, there appear to be plenty of live bacteria in the culture after the lysis step and before lyophilization.  The OD600 of the CBG culture increases from ~3 to ~5 in an hour!  If ghost cells are created after induction of lysis but not all bacteria are lysed, then the mice are subsequently being immunized with a mixture of bacterial ghost cells and bacteria that were killed during the lyophilzation process, which calls into question the entire premise of the paper. Figure 2B is convincing that the ghost preparation is not toxic to cells, but exactly what was used in this experiment?  Fresh ghosts or lyophilized?  What media was used to grow the Vero cells?  Too many details are lacking. The description of the ELISA assay is insufficient. What was the dilution of each type of sample?  This is a key point in interpreting the data.  End point dilution titers are superior to single dilution assay because it requires less interpretation.  When a single dilution assay is performed, it’s critically important to describe the assay in detail.  If the starting material in Fig 3A is undiluted serum, the result is rather different than if it was diluted 1:100.

Minor Comments:

Line 19 and throughout – “2LD50” should be “two LD50” Line 22 – “22MLD” should be “22 MLD” Line 47 – “proteinA” should be “protein A” Lines 47-48: Check wording and probably reword.  Do enteroviruses really encode an OmpA? Line 48 – “ovel” should be “novel” Line 56 – “37°C” should be “37 °C” (I dislike the additional space but it seems to have become standard.)  See also lines 91, 92, Line 73 – “Full-lengthopen” should be “Full-length open” Line 86 – “millilitres” should be ml Line 87 – “5ml” should be “5 ml” The authors appear to never include a space between the number and the abbreviation.  I think the space should be there because we wouldn’t write “fivemillitres” or “one hundredmicrograms.”  In any case, it’s not a substantive issue. Line 101 – “106Hep-2” should be “106 HEp-2” The cell line should be referred to as “HEp-2” throughout.  Capitalizing the E is definitely the correct way to state the name.  I’ve been corrected for this before myself. Line 116 – insert “our” or “the” before “institute.” Line 118 – “day14” should be “day 14” Line 130 – fix “OBGsand” Line 131 – Delete “And also,” Line 134 – fix “EVP1andCVP1” Line 154 – fix “lysisrate” Lines 159 – “1 x1010” should be “1x1010”  Correct spacing as needed throughout. The presentation of Fig 1 must be improved. In 1A the titles of each panel are not aligned to the same height.  1B doesn’t have a title.  The images in 1C could be re-ordered to align with Fig 1A.  The scale bars in 1C are difficult to see.  The numerical labels above each panel in D don’t have any obvious meaning and should be trimmed out.  I would also trim “FITC” from below each panel and have a single “FITC” and the bottom of 1C.    What is the between the upper left panel in 1C and the lower left panel?  Labeling is the same for both.  It should be immediately clear from the labels which antibody was used to stain which cells.  Consider placing the OBG controls along the top and the EBG/CBG on the bottom.  Let the left hand column be one antibody and the right column the other.  Place the antibody name above the column. Line 204 – typically people use symbols to denote statistically significant difference and not statistically insignificant (P>0.05) differences.

Author Response

Dear reviewer:

We thank your constructive criticisms and believe that we have addressed all of your concerns. We believe our findings have significance for those studying about a GB based vaccine against HFMD.

Sincerely,

Hui Wang

                             Response to Reviewer 2 Comments

Major Comments:

 1.You expressed your concern about “Lines 31-33. This sentence needs to be significantly revised.  What B cells secrete “serum-neutralization antibodies into the intestine”?  Additionally, the sentence confuses protective immunity with the effector immune response. MDV obviously generates and disseminated infection and is not restricted to the mucosa.  The immune mechanisms for clearing that infection from an infected individual are not necessarily the same as the correlate of protection against infection.  Lastly, the paper listed as a reference for this sentence has nothing to do with FMDV.”

Response 1: That is really a nice question. We revised this sentence, please see line 31-32. We also changed the reference, please see line 434.

2.You expressed your concern about “The authors often omit the space between a number and the word that follows it, and frequently between words. I’ve noted several instances below, but the paper deserves a careful review to fix all of these minor issues.  None of them alone is particularly noteworthy, but the frequency is great enough that it’s distracting.”

Response 2: Yes, we totally agree with you. We did fix all of these minor issues.

3.You expressed your concern about “The axes on 1A read “time post inducing” but I think this is really the time post temperature shift as stated in the legend. See also lines 89-92 in methods, which state that the cultures were induced at OD600=0.3 and temperature shifted at OD600=0.6.  The increases in OD600 seen after two hours is extremely concerning.  Do viable bacteria remain in the culture after induction of lysis?  Line 96 states that lyophilized samples were tested for viable cells, but based on this figure, there appear to be plenty of live bacteria in the culture after the lysis step and before lyophilization.  The OD600 of the CBG culture increases from ~3 to ~5 in an hour!  If ghost cells are created after induction of lysis but not all bacteria are lysed, then the mice are subsequently being immunized with a mixture of bacterial ghost cells and bacteria that were killed during the lyophilzation process, which calls into question the entire premise of the paper. Figure 2B is convincing that the ghost preparation is not toxic to cells, but exactly what was used in this experiment?  Fresh ghosts or lyophilized?  What media was used to grow the Vero cells?  Too many details are lacking.”

Response 3: Sorry to confuse you. We did not describe clearly about the method. We corrected labels of the figure and some statements. And also we added some details to the methods. Please see line 92-115 Yes, some viable bacteria still remain in the culture after induction of lysis. Repeated freezing and thawing method was used to get rid of live bacteria for the next step.

4.You expressed your concern about “The description of the ELISA assay is insufficient. What was the dilution of each type of sample?  This is a key point in interpreting the data.  End point dilution titers are superior to single dilution assay because it requires less interpretation.  When a single dilution assay is performed, it’s critically important to describe the assay in detail.  If the starting material in Fig 3A is undiluted serum, the result is rather different than if it was diluted 1:100.”

Response 4: We added more details to the method. Please see line 162-163. The first dilution is 1:10 in each assay, then made serial dilutions. The Fig 3A was displayed in antibody titers, but analyzed with log10 value.

Minor Comments:

1.You expressed your zconcern about “Line 19 and throughout – “2LD50” should be “two LD50” Line 22 – “20MLD” should be “20 MLD”. 

Response 1: Thank you for you good suggestions. We checked throughout the paper, and corrected all (17 parts)and labels of Figure 5 .

2.You expressed your concern about “Line 47 – “proteinA” should be “protein A”.”

Response 2: We corrected, thank you.

3.You expressed your concern about “Lines 47-48: Check wording and probably reword. Do enteroviruses really encode an OmpA?.”

Response 3: Sorry about this mistake, we deleted “virus and”. Please see line 52.

 4.You expressed your concern about “Line 48 – “ovel” should be “novel” Line 56 – “37°C” should be “37 °C” (I dislike the additional space but it seems to have become standard.)  See also lines 91, 92, Line 73 – “Full-lengthopen” should be “Full-length open” Line 86 – “millilitres” should be ml Line 87 – “5ml” should be “5 ml” The authors appear to never include a space between the number and the abbreviation.  I think the space should be there because we wouldn’t write “fivemillitres” or “one hundredmicrograms.”  In any case, it’s not a substantive issue.”

Response 4: We checked all the manuscript, and corrected one by one.

5.You expressed your concern about “Line 101 – “106Hep-2” should be “106 HEp-2” The cell line should be referred to as “HEp-2” throughout.  Capitalizing the E is definitely the correct way to state the name.  I’ve been corrected for this before myself.”

Response 5: We corrected, please see line 122,123, 206.

6.You expressed your concern about “Line 116 – insert “our” or “the” before “institute.”

Response 6: We inserted “our“.

7.You expressed your concern about “Line 118 – “day14” should be “day 14.”

Response 7: We corrected. Please see line 139.

8.You expressed your concern about “Line 130 – fix “OBGsand” Line 131 – Delete “And also.”

Response 8: We revised all according to your suggestions, please see line 151 and line 152.

9.You expressed your concern about “Line 134 – fix “EVP1andCVP1” Line 154 – fix “lysisrate”.”

Response 9: We fixed them, please see line 161 and line 182.

10.You expressed your concern about “Lines 159 – “1 x1010” should be “1x1010”. Correct spacing as needed throughout.”

Response 10: We corrected, please see line 187 and 188. We added spaces to the whole manuscript.

11.You expressed your concern about “The presentation of Fig 1 must be improved. In 1A the titles of each panel are not aligned to the same height.  1B doesn’t have a title.  The images in 1C could be re-ordered to align with Fig 1A.  The scale bars in 1C are difficult to see.  The numerical labels above each panel in D don’t have any obvious meaning and should be trimmed out.  I would also trim “FITC” from below each panel and have a single “FITC” and the bottom of 1C.    What is the between the upper left panel in 1C and the lower left panel?  Labeling is the same for both.  It should be immediately clear from the labels which antibody was used to stain which cells.  Consider placing the OBG controls along the top and the EBG/CBG on the bottom.  Let the left hand column be one antibody and the right column the other. Place the antibody name above the column.”

Response 11: We improved Figure 1. The scale bars in 1C were added in figure legend.

12.You expressed your concern about “Line 204 – typically people use symbols to denote statistically significant difference and not statistically insignificant (P>0.05) differences.”

Response 12: We corrected. Please see Figure 2 and Figure 3.

Round 2

Reviewer 2 Report

The paper has been revised and is much improved.  However, I'm still really bothered by the fact that much of the immunizing material consists of dead bacteria in addition to bacterial ghosts.  The methods give a clearer statement of what was done, so readers can decide for themselves the importance of this point.  

Major Comments:

1.  The description of the ELISA methods require further improvement.  I suggest insert a sentence at line 138-140 that specifically states what the end point dilution is.  For example, I would write something like "Serum samples were serially diluted in 10-fold dilutions from 1:10 to 1:100,000.  The endpoint dilution titer was calculated as the serum dilution resulting in an absorbance reading of 0.2 units above background."  As the manuscript is currently written, we don't know how the titers shown in the graphs were calculated.  Understanding how the titer was calculated is necessary for being able to interpret the results.  I'm sure this point can be easily fixed by the authors.

Minor Comments:

Line 59 - Go ahead and state which Vero and which HEp cell lines.  Later in the manuscript it's clear that HEp-2 cells were used, but I think it's best to put the complete name of each cell line in the methods section.  Also, there are different Vero cell lines, so if some specific sub-line was used, like Vero E6, please be sure to include. Line 97:  "survival" should be "surviving" Line 139 - "form" should be "from" Line 116 - "with Beijing Institute" should be "with the Beijing Institute" Line 118 - "institute" should be "institute's" Line 120 - "guide lines of institute" should be "guidelines of the institute" Line 347 - To be consistent, please italicize the journal name in reference #4.

Author Response

Dear reviewer:

We thank your good suggestions and believe that we have addressed all of your concerns. 

Sincerely,

Hui Wang

Professor of Beijing Department of Infection Immunity&Defense

State Key Laboratory of Pathogen and Biosecurity

Beijing Institute of Microbiology and Epidemiology, Beijing, China

20 Dongda Street, Fengtai District, Beijing 100071, China

E-mail:  geno0109@vip.sina.com

Tel: +86 10 66948532

Fax: +86 10 66948532

                                     Response to Reviewer 2 Comments

Major Comments:

 1.You expressed your concern about “The description of the ELISA methods require further improvement.  I suggest insert a sentence at line 138-140 that specifically states what the end point dilution is.  For example, I would write something like "Serum samples were serially diluted in 10-fold dilutions from 1:10 to 1:100,000.  The endpoint dilution titer was calculated as the serum dilution resulting in an absorbance reading of 0.2 units above background."  As the manuscript is currently written, we don't know how the titers shown in the graphs were calculated.  Understanding how the titer was calculated is necessary for being able to interpret the results.  I'm sure this point can be easily fixed by the authors.”

Response 1: Thank you for you good suggestions. We revised this section, please see line 166-168.

Minor Comments:

1.You expressed your concern about “Line 59 - Go ahead and state which Vero and which HEp cell lines.  Later in the manuscript it's clear that HEp-2 cells were used, but I think it's best to put the complete name of each cell line in the methods section.  Also, there are different Vero cell lines, so if some specific sub-line was used, like Vero E6, please be sure to include. ”. 

Response 1: Thank you for you good suggestions. We checked throughout the paper, and corrected all about cell line’s name.

2.You expressed your concern about “Line 97:  "survival" should be "surviving".”

Response 2: We corrected, thank you.

3.You expressed your concern about “Line 139 - "form" should be "from".”

Response 3: We corrected, thank you.

4. You expressed your concern about “Line 116 - "with Beijing Institute" should be "with the Beijing Institute".”

Response 4: We corrected, thank you.

5.You expressed your concern about “Line 118 - "institute" should be "institute's".”

Response 5: We corrected, thank you.

6.You expressed your concern about “Line 120 - "guide lines of institute" should be "guidelines of the institute".”

Response 6: We corrected, thank you.

7.You expressed your concern about “Line 347 - To be consistent, please italicize the journal name in reference #4.”

Response 7: We corrected, thank you.
